# Alpha-Germanium Nanolayers for High-Performance Li-ion Batteries

**DOI:** 10.3390/nano12213760

**Published:** 2022-10-26

**Authors:** Laura Sierra, Carlos Gibaja, Iñigo Torres, Elena Salagre, Juan Ramón Avilés Moreno, Enrique G. Michel, Pilar Ocón, Félix Zamora

**Affiliations:** 1Departamento de Química Física Aplicada, Universidad Autónoma de Madrid, 28049 Madrid, Spain; 2Departamento de Química Inorgánica, Universidad Autónoma de Madrid, 28049 Madrid, Spain; 3Departamento de Física de la Materia Condensada, Universidad Autónoma de Madrid, 28049 Madrid, Spain; 4Condensed Matter Physics Center (IFIMAC), Universidad Autónoma de Madrid, 28049 Madrid, Spain; 5IMDEA-Nanociencia, Ciudad Universitaria de Cantoblanco, 28049 Madrid, Spain

**Keywords:** germanium, Li-ion batteries, anode material, 2D nanolayers, energy storage

## Abstract

The exfoliation of tridimensional crystal structures has recently been considered a new source of bidimensional materials. The new approach offers the possibility of dramatically enlarging the library of bidimensional materials, but the number of nanolayers produced so far is still limited. Here, we report for the first time the use of a new type of material, α-germanium nanolayers (2D α-Ge). The 2D α-Ge is obtained by exfoliating crystals of α-germanium in a simple one-step procedure assisted by wet ball-milling (gram-scale fabrication). The α-germanium nanolayers have been tested as anode material for high-performance LIBs. The results show excellent performance in semi-cell configuration with a high specific capacity of 1630 mAh g^−1^ for mass loading of 1 mg cm^−2^ at 0.1 C. The semi-cell was characterized by a constant current rate of 0.5 C during 400 cycles and different scan rates (0.1 C, 0.5 C, and 1 C). Interestingly, the structural characterization, including Raman spectroscopy, XRPD, and XPS, concludes that 2D α-Ge largely retains its crystallinity after continuous cycling. These results can be used to potentially apply these novel 2D germanium nanolayers to high-performance Li-ion batteries.

## 1. Introduction

Since the discovery of graphene in 2004 [1], new 2D materials have become strategic targets due to their unusual (opto)electronic properties and potential technological impact [2]. In the last decades, the scientific community has been focused on obtaining a wide variety of 2D materials that can be prepared by bottom-up methods, such as chemical synthesis or vapor growth, as well as by top-down methods, such as micromechanical exfoliation, electrochemical exfoliation, and/or liquid phase exfoliation (LPE) [3] based on van der Waals crystals exfoliation. As a consequence, different families, such as transition metals dichalcogenides (TMDs) [4], layered double hydroxides (LDHs) [5], Mxenes [6], and Xenes [7] have appeared, among others. Among all these techniques, LPE is one of the most used methods to produce 2D nanosheets on a large scale. The search for 2D materials has gone beyond these layered materials, exploring the direct exfoliation of nonlayered 3D crystals using LPE [8]. In 2017, Guan et al. [9] were able to exfoliate monoclinic tungsten trioxide crystals by LPE, opening the door to use other nonlayered strongly bonded crystals as starting materials to prepare 2D nanosheets following this procedure. Thus, our group has reported, very recently, a method to prepare α-germanium nanolayers (2D α-Ge) [10]. However, these new 2D materials are still in their infancy; the expected structures will present attractive grain boundaries with bond saturations that will affect their physical and chemical properties and potential applications.

In general, 2D materials show unprecedently excellent properties in the energy storage field compared with their bulk counterparts due to their increase in the amount of active sites to accommodate or reach guest ions facilitating the Li^+^/Na^+^ transport, which notably increases the rate performance. Moreover, the use of 2D nanosheets obtained by LPE represents a high potential in this field, with different applications, such as photocatalysis [11], electrode materials for supercapacitors [12], or active material for LIBs [13,14]. Indeed, the increasing demand for renewable energy resources has pushed the development of new devices for high-performance LIBs [15]. Nowadays, commercial LIBs are mainly conformed with graphite as electrode material in the anode, which confers a low theoretical capacity of 372 mAh g^−1^ [16]. For this reason, anodes involving alloy reactions, such as Si [17,18], Ge [19], or Sb [20], have garnered much attention in particular, considering their theoretical capacities of 3579 mAh g^−1^, 1600 mAh g^−1^, and 660 mAh g^−1^, respectively, for LIBs. Additionally, anodes involving alloy reactions have begun to be used for potassium-ion and sodium-ion batteries (PIBs/ SIBs) with Ge [21] and Sn [22]. Silicon is the first candidate due to its large theoretical specific capacity of 3579 mAh g^−1^ for Li_15_Si_4_ and its low cost compared to other metals. The limitation of silicon as an anode in LIBs is its poor electrical conductivity and fragility after the lithiation process and, consequently, a high loss of capacity during cyclability [23]. The Ge-based anodes offer an attractive alternative because of (i) its electrical conductivity, which is 100 times higher than Si; (ii) the lithium-ion diffusivity, 400 times higher than Si; and (iii) the toughness of Ge-particles after the cyclability, which increases the batteries’ life [24]. Despite these better properties, Ge-based anodes suffer volume changes during successive lithiation/delithiation cycles that induce structural deformation and, consequently, a loss of specific capacity. Therefore, the main research strategies are focused on Ge nanostructures to increase the surface area, displaying good results for nanoparticles [25] and nanowires [26]. This increment of the surface area mitigates the volume expansion and improves lithium-ion diffusion. However, to obtain these nanostructures of germanium, it is necessary to incorporate carbon material during the synthesis to provide a size reduction of active materials and a suitable dispersion into the carbon matrix, increasing the cost and complexity of the procedure. Several works have investigated structural alternatives to provide more stability to Ge as anode material without the use of a carbon matrix, i.e., (i) porous Ge [27,28]; (ii) Ge Quantum Dots [29]; and (iii) a combination of Ge and GeO_2_ to provide more specific capacity [28]. The main disadvantage of those materials is the additional effort in the material synthesis, which increases the cost of the manufactured anode and the scaling up for industrial use. In this way, recent investigations focus on alternatives to obtain materials using cost-effective and large-scalable methods [8].

Recently, a new method for a gram-scale production of 2D α-Ge by LPE has been reported by us [10]. Herein, we have studied, for the first time, the potential application of this new 2D material as an anode for LIBs. The 2D α-Ge modified electrodes show an excellent specific capacity of 1104 mA h g^−1^ at 0.5 C. Additionally, a deep characterization before and after the battery tests has been carried out, obtaining relevant information about the changes in the morphology and structure of 2D α-Ge after the charge/discharge processes, which confirms high crystallinity retention of 2D α-Ge structures.

## 2. Materials and Methods

### 2.1. Synthesis of 2D α-Ge

The preparation of 2D α-Ge was carried out as previously reported by us [10]. A 20 mL ball-milling reactor was charged with 1 g of bulk α-germanium crystals (99.999%, purity), 30 stainless steel balls, and 1 mL of a mixture 2-propanol: H_2_O (4:1 *v*/*v*). The reactor was then stirred for 60 min at 3000 rpm. Then, the wet powder was dried under a vacuum for 24 h at 60 °C and stored under an Ar atmosphere.

All reagents and solvents purchased were used without further purification. Lithium hexafluorophosphate solution in ethylene carbonate (EC) and diethyl carbonate (DEC) (50:50 *v*/*v*), and vinylene carbonate (VC) were purchased from Sigma Aldrich (St. Louis, MO, USA), and α-germanium crystals (99.999% purity) were purchased from Smart Elements Co. (Wien, Austria).

### 2.2. Electrochemical Measurements

For 2D α-Ge, a 2032-coin cell was assembled in an Ar-filled glovebox. Different amounts of 2D α-Ge were used as working electrodes (1, 2, 5, and 10 mg cm^−2^). The active material was incorporated as slurry. VGCF was used as a conductive agent with CMC as a binder. The weight ratio of Ge:VGCF:CMC was 80:10:10, respectively. The mixture was cast onto a current copper collector with a doctor blade and dried overnight at 60 °C under a vacuum. After drying, the electrode was cut into a disk measuring 12 mm in diameter. The thickness of electrodes was 10 and 17 µm for 1 and 2 mg cm^−2^, respectively. Lithium metal was used as the counter and reference electrode, and Whatman glass fiber (GF/C) as the separator. The electrolyte was LiPF_6_ 1M in ethylene carbonate (EC) and diethyl carbonate (DEC) 50:50 and 10% vinyl carbonate. The measurement was performed using biologic multichannel potentiostatic-galvanostatic with impedance module BSC-815. Cyclic voltammetry at a scan rate of 0.1 mV·s^−1^ and galvanostatic charge/discharge (GCD) were performed between 1.8 to 0.02 V at 0.1 C, 0.5 C, and 1 C, with 1 C = 1600 mAh g^−1^. Electrochemical impedance spectroscopy (EIS) measurements were carried out between 10 kHz and 0.01 Hz with an amplitude of 10 mV at open circuit potential (OCP). Zview^®^ software from Scribner for EIS analyses was used. The specific capacities were calculated concerning the mass of Ge.

Powder X-ray diffraction (XRPD) measurements were carried out on a Bruker D8 (Bruker, Billerica, MA, USA) with CuKα radiation with a rapid detector (lynxeye).

Raman spectra were recorded with a BWS415 i-Raman coupled to a BAC151B microscope equipped with 20× and 50× objectives and a 532 nm laser as excitation light. All the spectra were recorded with the 50× objective, laser powers from 0.70 to 35 mW, and integration times between 5000 and 20,000 ms.

TEM was recorded in the JEOL JEM 3000F TEM system (JEOL, Tokyo, Japan) with an accelerating voltage of 300 kV, and SEM images were obtained on a Philips XL 30 S-FEG microscope operating at an accelerating voltage of 10 kV.

XPS core-level spectra were obtained using Mg Kα (1253.6 eV) and Al Kα (1486.6 eV), and a Phoibos 150 electron analyzer whose axis coincided with the surface normal. The core-level binding energies were calibrated using the binding energies of C 1s and Au 4f in contact with the sample as references. The line shape of core levels was fitted using a Shirley background and asymmetric singlet pseudo-Voigt functions. The fit was optimized using a Levenberg–Marquardt algorithm with a routine running in IGOR Pro (WaveMatrix Inc., Lake Oswego, OR, USA) [30]. The fit quality was judged from a reliability factor, the normalized χ^2^.

## 3. Results and Discussion

### 3.1. Preparation and Characterization of a-Germanium Nanolayers

The morphology and microstructure of the 2D α-Ge were investigated by transmission electron microscopy (TEM) and scanning electron microscopy (SEM). Figure 1a shows a typical TEM image of a 2D α-Ge with lateral sizes close to 1–2 µm. Moreover, TEM images have been used to evaluate the size distribution of the 2D α-Ge (Appendix A), confirming that most of the nanolayers present sizes between 0.5 and 3 µm. SEM images (Figure 1b and Appendix A) of characteristic 2D α-Ge confirm the size of the nanolayers that were also confirmed by TEM. The atomic force microscopy (AFM) images reveal typical 2D α-Ge with the characteristic bidimensional morphology, displaying thicknesses between 10 and 30 nm and lateral dimensions around 3 µm (Figure 1c, Appendix A).

Figure 1d shows the diffraction patterns of the powder 2D α-Ge. The presence of the characteristic peaks of crystalline Ge (JCPDS card no. 03-065-0333) at 2θ 27.20°, 45.24°, 53.64°, and 65.99° correspond to the (110), (220), (311), and (400) planes of the diamond structure, respectively. The calculated lattice constant is *a* = 5.658 Å for a cubic phase Ge. Characteristic diffraction peaks of hexagonal GeO_2_ (JCPDS card No. 83-0543) are not observed in 2D α-Ge. Hence, the exfoliation procedure does not produce oxidation or changes in the structure of the α-Ge crystals. Raman spectra of 2D α-Ge were measured with a laser power of 1.75 mW, showing a characteristic peak at 298 cm^−1^, which is assigned to crystalline Ge. We can rule out the presence of hexagonal GeO_2_ in the sample because no characteristic peaks of crystalline GeO_2_ at 212, 261, and 440 cm^−1^ are observed [31]. Furthermore, we did not observe the presence of peaks corresponding to other Ge oxide derivatives, such as tetrahedral GeO_4_ [32].

The Raman spectrum is essential to correctly identify the characteristic features of crystalline Ge (298 cm^−1^), amorphous Ge (280–270 cm^−1^), and the presence of potential contaminants as different Ge oxides. We detected a high dependence on the Raman spectra with applied laser power. At higher laser powers >7 mW, we observed two effects: (i) crystalline Ge peak at (298 cm^−1^) shifts to lower wavenumbers (redshift), probably due to the amorphization of the sample and the generation of different Ge oxides; (ii) in situ generation of Ge oxides induced by the laser action (Appendix A for additional information). At high laser power, the observed peaks can be assigned to normal modes of α-GeO_2_ at 122, 166, 212, 443, and 590 cm^−1^ [33]. In addition, a new band around 312 cm^−1^ corresponding to a bending mode of GeO_4_ is observed [32]. However, low laser powers ensure the correct analysis of the material, and we obtained similar characterizations working at 1.75 mW (Figure 1e) and at 3.5 mW (Appendix A).

It is worth noting that those Ge oxides are not present in the sample, but are in situ generated by the laser energy. Then, we fixed 7 mW as a cut-off for analysis purposes. The laser exposure effect will influence the analysis at higher power, reaching partially wrong conclusions.

We used X-ray photoelectron spectroscopy (XPS) to analyze 2D α-Ge anode materials before and after cycling. Figure 1f collects the Ge 3d and O 2s core levels for the pristine anodes before cycling. The shown binding energy region includes Ge 3d (approx. 30–35 eV BE) and O 2s (approx. 23–25 eV). Ge 3d presents two prominent components at 29.7 eV and 33.0 eV BE, which are identified as elemental Ge (unoxidized) and Ge^4+^ (GeO_2_), respectively [34]. After deconvolution of the line shape, an additional suboxide component at 31.5 eV corresponding to Ge^2+^ (GeO) is identified. Less intense components are described in the Appendix A. We analyzed the intensities of the Ge components, considering the electron mean free path and the change in the relative intensities when using different photon energies, 1253.6 eV and 1486.6 eV (Appendix A).

We conclude that the thickness of the Ge^4+^ layer is (7 ± 0.7) nm for 2D α-Ge anodes. The appearance of Ge^4+^, Ge^2+^, and Ge suggests the presence of surface oxidation for 2D α-Ge anodes, either due to air exposure before XPS analysis or to preparation treatments of the anode. The Ge^4+^ layer is thicker than similar samples analyzed before [10] due to prolonged exposure to ambient conditions and the incorporation into a carbon matrix. It is important to remark that XPS is extremely sensitive to surface composition. Thus, different Ge oxides can be confirmed on the surface, but not inside the electrode.

### 3.2. Electrochemical Characterization

The electrochemical performance of 2D α-Ge anodes was evaluated by cyclic voltammetry (CV), galvanostatic charge/discharge cycling (GCD), and electrochemical impedance spectroscopy (EIS). Figure 2a shows the cyclic voltammetry of 2D α-Ge with 1 mg cm^−2^ in the potential range of 1.5 to 0.02 V vs. Li/Li^+^ at a scan rate of 0.1 mV s^−1^ for the first three scans.

In the first one (cathodic direction), a broad redox peak located at 0.11–0.14 V corresponds to lithium insertion and the formation of different Ge alloys. In the anodic scan, a peak located at 0.58 V corresponds to the de-alloying process of Ge_x_-Li composites. In the second and third scans, the cathodic peak at 0.11–0.14 V disappeared. This is probably due to the solid electrolyte interface (SEI) formation from the electrolyte/electrode reaction.

In contrast, three new cathodic broad peaks emerge. A shift of Li insertion to more positive potential values suggests a multi-step lithium insertion mechanism as the electrochemical reaction of Ge with lithium. The presence of a more significant number of peaks at (0.52 0.36, 0.16, and 0.07 V) corresponds to different alloys Ge_x_Li (Li_9_Ge_4_, Li_7_Ge_4_ Li_15_Ge_4_) reported in the literature [35,36]. In the anodic scan, we observe the same peak at approximately 0.58 V, but it slightly shifts when the cycle number increases. The changes in the structure of crystalline germanium were one of the reasons for its performance [37]. Appendix A shows the first five scans of 2D α-Ge with 1 mg cm^−2^. After the third scan, the intensity loss in the alloy/de-alloy process is lower than in the first cycles, achieving good stability and suggesting excellent electrochemical reversibility. Previous *operando* XRD and XAS studies indicate that the reaction mechanism of lithiation involves two steps [38]. During the first one, crystalline Ge forms amorphous Li_9_Ge_4_ in the beginning, and then, the conversion of the remaining crystalline Ge goes into amorphous Ge. After that, the formation of the heterogeneous amorphous phase represented by Li_x_Ge alloys is finally transformed into the crystalline Li_15_Ge_4_ alloy. During the delithiation process, the crystalline Li_15_Ge_4_ alloy is transformed into amorphous Li_x_Ge alloys and eventually forms amorphous Ge.

The lithium storage performance of the as-prepared 2D α-Ge samples as an anode was evaluated by galvanostatic charge/discharge (GCD). The results show a good correlation with the CV data; therefore, for the initial cycle, there is a discharge voltage plateau at ca. 0.5 V, which was assigned to the formation of the Li_x_Ge alloy.

The GCDs were performed with cut-off potentials 0.02 and 1.80 V. In all cases, two initial cycles at 0.05C were carried out to obtain a better performance in the SEI because of the high reactivity of the electrolyte in the first cycles [39]. CE% is lower than 25% in those cycles, attributable to the SEI formation (Appendix A).

Figure 2b shows the GCD results at 0.5 C for 2D α-Ge anodes. The capacity retention was evaluated for over 400 charge/discharge cycles. We observe that the first cycle for the electrode with higher mass loading (2 mg cm^−2^) shows a higher specific capacity (1526 mAh g^−1^) in comparison to 1 mg cm^−2^ (1104 mAh g^−1^). After 20 cycles, the performance becomes similar for both electrodes, indicating some relationship between mass and lithiation/delithiation reactions. In both materials, upon cycling, the capacity drops in comparison to the initial state, and afterward, the electrode with 1 mg cm^−2^ presents better stability. In fact, after 110 cycles, this electrode has 711 mAh g^−1^ vs. 566 mAh g^−1^ for the 2 mg cm^−2^ material. Finally, after 200 cycles, the electrode with 1 mg cm^−2^ has a specific capacity of 608 mAh g^−1^ vs. 368 mAh g^−1^ for the electrode with 2 mg cm^−2^.

The significant loss of capacity occurs in the first 50 cycles, with 34% for electrodes with 1 mg cm^−2^ and 43% for electrodes with 2 mg cm^−2^. Additionally, over 200 cycles, the electrode with 2 mg cm^−2^ shows a dramatic capacity loss of 75% compared to the initial value; meanwhile, for 1 mg cm^−2^, there was only a loss of 45% compared to the initial value. This behavior can be related to the amount of Ge per unit area of each electrode. The capacity fading is related to a significant volume change after the cycling of the alloy/de-alloying process. This fact will be more significant with the increased thickness of the active layer. Table 1 shows the results corresponding to the 400 cycles at 0.5 C, and Appendix A shows different scan rates for 2D α-Ge with 1 and 2 mg cm^−2^.

Tentative experiments with high-mass loading electrodes (5–10 mg cm^−2^) were carried out (Figure 2c). The main purpose is to obtain more capacity and determine where the upper limit is in terms of active material amount. The literature shows several studies with amounts ranging between 1 and 8 mg cm^−2^ of active material, and the conclusion is clear: the specific capacity decreases with increasing mass loading [29]. Figure 2c shows how the performance decreases with increasing mass loading, displaying a low specific capacity of 254 mAh g^−1^ and 85 mAh g^−1^ at cycle 10, and 50% capacity loss after 40 cycles, and about 80% with 5 mg cm^−2^ and 10 mg cm^−2^, respectively. This fact confirms that the use of electrodes with highly dispersed material and, therefore, with a low amount of active material results in better performance [29].

The capacity retention of Ge has been measured at different charge/discharge rates (0.1 C, 0.5 C, and 1 C) for 2D-α-Ge anodes. The comparison between 2D α-Ge with 1 mg cm^−2^ and 2 mg cm^−2^ and the specific discharge capacity values are included in Table 1 and Figure 3a,b and Appendix A. The capacity loss at 0.1 C and 0.5 C is generally similar for both electrodes. The main loss is observed when the current is increased to 1 C, and the capacity is about half the initial value. To evaluate the material stability, we return to 0.1 C, and the conclusion is clear: the 2D α-Ge with 1 mg cm^−2^ retains the main capacity with a 5.5% loss of ca. being 13.3% for 2D α-Ge with 2 mg cm^−2^. The GCD analysis shows a better performance for 2D α-Ge with a mass loading of 1 mg cm^−2^ for long cycling at a constant current rate and different scan rate. The excellent performance can be attributed to the stability of the material during the alloy/de-alloy process due to the crystallinity retention after the cycling, which provides more capacity and increases the useful life. Regarding the Coulombic efficiency (CE%), after the first 5–10 cycles, around 100% is achieved, probably due to the SEI formation/stabilization. At (0.5 C) and a mass load of 2 mg cm^−2^, 100% of CE is found; meanwhile, for 1 mg cm^−2^ this value is near 95%. For 0.1 C, the CE% values are lower than 95% in the initial cycles for 1 mg cm^−2^, decreasing greatly with 2 mg cm^−2^, but for the last cycles, when it turns to 0.1 C, CE% values are close to 100% in both cases.

Additionally, a half-cell with bulk Ge as an anode active material was evaluated under the same conditions as the 2D α-Ge material (Figure 3c). The results show a specific capacity of 353 mAh g^−1^ at cycle 100, which remains constant up to about cycle 400, significantly lower than that obtained by 2D α-Ge (697 mAh g^−1^).

Electrochemical impedance spectroscopy (EIS) characterization was carried out to get new insights into the charge transport kinetics of Ge electrodes and to understand the resistance characteristics of these electrodes. Figure 4a shows the Nyquist plot for 2D α-Ge 1 mg cm^−2^ after 10, 100, 200, and 400 cycles at a scan rate of 0.5 C. Appendix A shows similar plots at different C-rates. The plot shows a semicircle in the high–medium frequency (10^5^–40 Hz) and an oblique line in the low-frequency region. The first intercept with the real axis (in high frequency) represents the resistance contribution of the electrolyte and all the contributions to the resistance of the internal and external contacts (R_s_). The second intercept, the diameter of the semicircle (in high–medium frequency), can be associated with the charge transfer R_ct_ from the electrode/electrolyte interface. Finally, the line in the low-frequency region can be attributed to the resistance of Li^+^ diffusion within the electrode, in line with what has been described by He Xia et al. [40].

The Nyquist plots were fitted based on the equivalent circuit R_s_ (CPE_1_//R_ct_) W_o_ (Figure 4a). It considers R_s_ as the electrolyte ohmic resistance in a semi-cell in series with a (CPE_1_//R_ct_ ) representing the resistance of the charge transfer process. R_ct_ is associated with the active material, and CPE_1_ is the double-layer capacitance of the electrode. Both variables depend on the solid electrolyte interface (SEI) formation and the changes in the electrode (e.g., changes in surface area and surface morphology). A line for low frequencies at approximately 45° corresponds to a Warburg impedance element and provides information about the Li^+^ ionic diffusion into Ge-based electrodes. The values obtained after the fitting process based on the proposed circuit are included in Appendix A.

The fitting parameters (R_s_, R_ct_, and W_0_) provide information about the changes that occur during the cycling process. In that sense, for constant cycling (400 cycles) at 0.5 C, for 2D α-Ge with 1 mg cm^−2^, significant changes are found after 200 cycles. The R_ct_ value is 43.8 Ω, slightly lower than the value after 10 cycles (50.5 Ω), but the W_o_ parameters (W_o_-R, W_o_-T, and W_o_-P) change notably. Furthermore, after 400 cycles, the R_ct_ increases until 94.8 Ω, and the W_o_ continues with a similar tendency as observed after 200 cycles. The R_s_ values are slightly lower after 200 and 400 cycles, and the CPE_1_ values corresponding to the double-layer capacitance of the electrode do not experiment with significative changes (1.90–2.00 × 10^−4^ F).

EIS parameters at different scan rates are also evaluated for 2D α-Ge 1 and 2 mg cm^−2^. The changes are lower than observed in continuous mode, but the *R_ct_* value generally increases after 10 cycles (0.5 C) and after 10 cycles (1 C). Regarding W_o_ parameters, the fitting data show a similar tendency for 1 mg cm^−2^. There is no clear trend in the case of 2D α-Ge 2 mg cm^−2^, but there are many changes during the process, demonstrating that the material needs more cycles to stabilize.

Furthermore, an estimation of Li^+^ diffusion coefficients (*D_Li_*^+^) was obtained from the analysis of the relationship between *Z_real_* and the angular frequency (*ω*) components of the Warburg element in the diffusional region (low-frequency region) for different operation modes. This evaluation is based on Fick’s laws for a semi-infinite linear diffusion model and is calculated by Equations (1) and (2) [41].
(1)$$Z_{real}=R_{CT}+R_{s}+\sigma \omega^{-1/2}\;$$
(2)$$D_{Li^{+}}=\frac{R^{2}T^{2}}{2A^{2}n^{4}F^{4}c^{2}\sigma^{2}}$$
where *R* is the ideal gas constant (8.314 J K^−1^ mol^−1^), *T* is the temperature (298 K), *A* is the electrode area (1.2 cm^2^), *n* is the number of electrons transferred per molecule during the reaction (*n* = 1), *c* is the concentration of Li^+^ in the bulk electrode, *F* is the Faraday constant (96,485 C mol^−1^), and *σ* is the Warburg coefficient.

The Warburg coefficient is obtained from the linear slope of the graphical representation of *Z_real_* versus ω^−1/2^ (Equation (1)) in the mid–low frequency region (Figure 4b).

Those values introduced in Equation (2) provide *D_Li_*^+^ for each measurement. Figure 4c shows a plot with *D_Li_*^+^ (filled columns) and R_ct_ (dashed columns) for 2D α-1 mg cm^−2^ at 0.5 C during 400 cycles. The *D_Li_*^+^ value increases during the cycling process. After 10 cycles, the value was 1.02 × 10^−13^ cm^2^ s^−1^, while after 400 cycles, *D_Li_*^+^ go to 2.60 × 10^−12^ cm^2^ s^−1^. The *D_Li_*^+^ was also evaluated at different scan rates (Appendix A). In this case, the behavior is similar to the above result, increasing *D_Li_*^+^ from 6.90 × 10^−13^ cm^2^ s^−1^ after 10 cycles at 0.1 C to 3.20 × 10^−12^ cm^2^ s^−1^ after 10 cycles at 1 C. We conclude that 2D α-Ge 1 mg cm^−2^ increases the *D_Li_*^+^ during cycling.

All results for the rest of the experiments are collected in Figure 4c and Appendix A. There is a compromise between the charge transfer resistance and the Li^+^ ion diffusion since it is observed that as the number of cycles increases, Li-ion diffusion increases as well, but the resistance to charge transfer becomes higher. Therefore, there is a trade-off two between the two variables and the battery performance with high-capacity values and good cyclability without too many losses.

### 3.3. Physico-Chemical Characterization and Structural Analysis for Cycled Materials

After cycling in continuous mode, XRPD, Raman, XPS, and SEM were measured and compared with the materials before the cycling process. In this way, we could explain structural changes observed in the fail-mode operation for each material in the semi-cell.

Figure 4a shows the XRPD spectra of 2D α-Ge after 400 cycles at 0.5 C. The peaks at 2θ 27.20°, 45.24°, 53.64°, and 65.99° corresponding to 2D α-Ge are observed. This indicates that the electrode retains some of its crystallinity, and not all Ge turns amorphous during the lithiation/delithiation process.

Moreover, a Raman study before and after the cycling process can identify the ratio of amorphous and crystalline Ge. The peaks corresponding to crystalline Ge (298 cm^−1^) and amorphous Ge (270 cm^−1^ and 280 cm^−1^) are analyzed [42]. For 2D α-Ge anodes, we selected 1.75 mW as laser power. Figure 1e shows the Raman spectrum for 2D α-Ge pristine, and Figure 5b shows the Raman spectrum for 2D α-Ge after cycling. Deconvolution was performed to obtain the crystalline and amorphous contribution. Before and after cycling, we obtained between 75 to 80% of the crystalline contribution, and after cycling, the results are very similar to the initial value. The same behavior is obtained at 3.50 mW of laser power (Appendix A).

Previously, Zhou et al. demonstrated the influence of particle size in alloy reaction with Ge via SEM and TEM measurements. In the case of µm Ge particles, they suggest that two different reactions can occur during the lithiation process. The first reaction represents a surface reaction, and the second reaction represents the lithiation of the bulk structure [43]. In contrast, an amorphous core shell is formed during the lithiation process for Ge nanoparticles [44].

Concerning XPS measurements, Figure 5c shows the Ge 3d and O 2s core levels corresponding to 2D α-Ge anodes after the cycling process. The overall intensity of the Ge 3d–O 2s region is significantly reduced compared to the pristine anodes. The reduction is due to attenuation from an electrolyte layer still present in the analyzed samples, as evidenced by the observation of P 2p and F 1s core levels (Appendix A). We estimate a thickness of (3 ± 0.3) nm for the residual electrolyte layer. Besides the intensity attenuation, the most crucial change after cycling is removing the Ge oxides layer and the appearance of a new Ge component at 28.6 eV BE. This value is smaller than the BE of elemental Ge, and thus, it corresponds to Ge in a more reduced state. We identify this component with Ge^−^ and explain its origin by forming a layer of Li_x_Ge composition. The observation of a Ge^−^ component in a Li_x_Ge compound agrees with a similar finding for Li_15_Si_4_ [45]_._ A minor Ge^2+^ component is still detected in cycled anodes, indicating the presence of some remaining oxidized Ge. Considering the relative intensities of Ge^2+^ and Ge^−^, we estimate a thickness of the GeLi_x_ layer of (7 ± 1) nm for 2D α-Ge. The value of the Ge:Li stoichiometry in this layer cannot be accurately determined, as part of the Li 1s signal is coming from other sources (e.g., the remaining electrolyte layer), and it is difficult to distinguish from Li atoms in the GeLi_x_ layer.

Therefore, the presence of Ge oxides is limited to the first few atomic layers and is due to a surface oxidation process. This is evidenced by observing a sequence of GeO_2_, Ge suboxides, and elemental Ge, typical of surface oxidation of Ge, and by removing the Ge oxides after cycling the electrodes. The thickness of this layer is larger than the probing depth of XPS using Mg Kα (approx. 10 nm), as the Ge layer underneath is not observed.

Additionally, the morphology changes in cycled material were investigated by SEM for 2D α-Ge material (Appendix A). The images suggest that once subjected to the cycling process, the flakes suffer fractures induced by the lithiation process and volume expansion.

It is worth remarking that all the physico-chemical characterizations before and after the cycling process reveal similar and complementary conclusions, considering the specificity and sensitivity of each technique. For instance, XPS is much more sensitive to the surface composition than Raman, which is more sensitive to the average structure, or XRD, which is sensitive to the crystalline fraction of the bulk only. The presence of Ge oxides is not relevant and is related to the surface only, leading to the formation of a thin oxide layer. Finally, this new type of material, i.e., 2D α-Ge material, shows excellent performance as an anode in LIBs, mainly due to the increase in the active surface and the low amorphization during the cycling process.

## 4. Conclusions

In summary, we have evaluated the use of a novel 2D material based on α-Ge crystals that can be produced on a gram-scale as anode material for high-performance LIBs. The lithiation process is associated with high specific capacity values as 1630 mAh g^−1^ for 1 mg cm^−2^ mass loading at 0.1 C. Additionally, the study of mass loading with the 2D α-Ge material shows that the best performance is found for 1 mg cm^−2^ with high-capacity retention after 400 cycles, 599 mA h g^−1^ at 0.5 C. Further, the EIS measurements show an increase in DLi^+^ during the charge/discharge process, displaying values of 2.60 × 10^−12^ cm^2^ s^−1^ after 400 cycles. The physico-chemical analysis confirms high crystallinity retention of 2D α-Ge nanoflakes after continuous cycling.

Future studies with DFTs and/or molecular dynamics calculations could help elucidate the lithiation/delithiation reaction for 2D α-Ge structures. These results are promising for fabricating LIBs using 2D α-Ge as an anode material.

## Figures and Tables

**Figure 1 nanomaterials-12-03760-f001:**
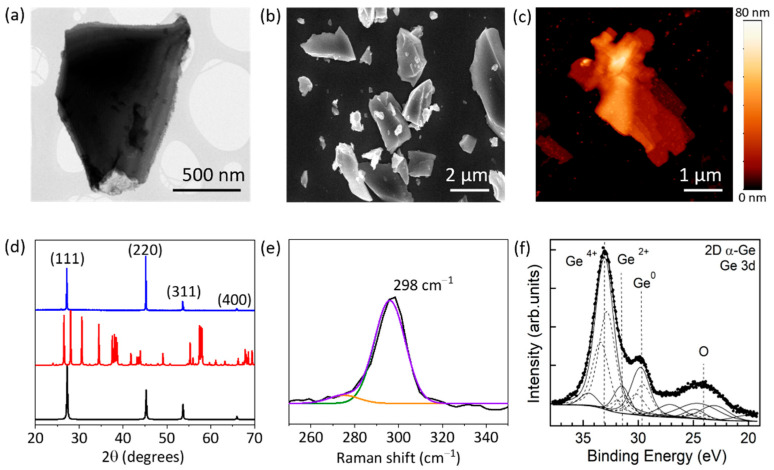
(**a**) TEM image of a typical 2D α-Ge. (**b**) General SEM image of characteristic 2D α-Ge. (**c**) AFM image of a typical 2D α-Ge. (**d**) XRPD patterns of 2D α-Ge (blue line), simulated hexagonal crystalline GeO_2_ (red line), and simulated crystalline Ge (black line). (**e**) Raman spectrum of 2D α-Ge before the cycling process. Raman spectrum measurement (black line), deconvoluted contributions of crystalline α-Ge at 298 cm^−1^ (green line), amorphous Ge at 275 cm^−1^ (orange line), and cumulative plot of both contributions (purple), laser power of 1.75 mW. (**f**) Ge 3d and O 2s XPS peaks (hν = 1253.6 eV) for 2D α-Ge before cycling. Experimental data (dotted line) and resulting fit (thick continuous lines). Thin lines correspond to the different contributions, and dashed lines to the spin-orbit splitting of Ge 3d.

**Figure 2 nanomaterials-12-03760-f002:**
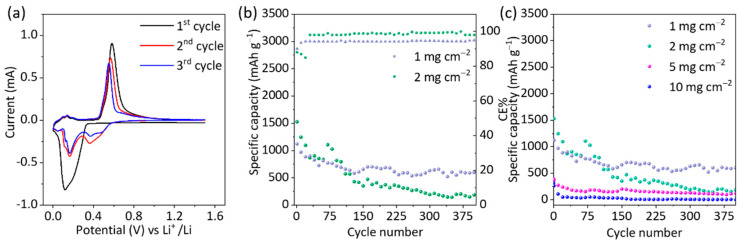
(**a**) CV of 2D α-Ge 1 mg cm^−2^ semi-cell at 0.1 mV s^−1^, 25 °C, first three cycles. (**b**) GCD at constant current rate 0.5 C over 400 cycles for 2D α-Ge 1 mg cm^−2^ semi-cell (purple dots) and 2D α-Ge 2 mg cm^−2^ semi-cell (green dots). (**c**) GCD at constant current rate 0.5 C over 400 cycles for 2D α-Ge 1 mg cm^−2^ semi-cell (purple dots), 2D α-Ge 2 mg cm^−2^ semi-cell (green dots), 2D α-Ge 5 mg cm^−2^ semi-cell (pink dots) and 2D α-Ge 10 mg cm^−2^ semi-cell (blue dots).

**Figure 3 nanomaterials-12-03760-f003:**
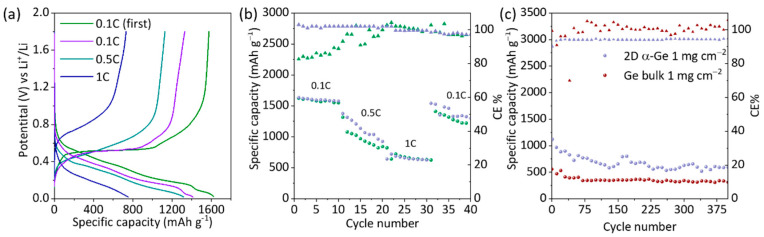
(**a**) GCD profiles of 2D α-Ge 1 mg cm^−2^ semi-cell at different scan rates. (**b**) GCD at different scan rates for 2D α-Ge Ge 1 mg cm^−2^ semi-cell (purple dots) and 2D α-Ge 2 mg cm^−2^ semi-cell (green dots). The triangles indicate the corresponding CE%. (**c**) GCD at constant current rate of 0.5 C over 400 cycles for 2D α-Ge 1 mg cm^−2^ semi-cell (purple dots) and Ge bulk 1 mg cm^−2^ semi-cell (dark red dots). The triangles indicate the corresponding CE%.

**Figure 4 nanomaterials-12-03760-f004:**
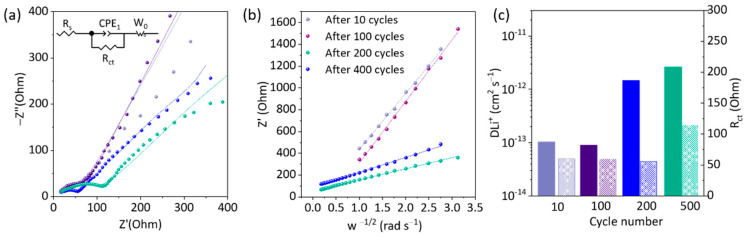
(**a**) Nyquist plot of 2D α-Ge 1 mg cm^−2^ semi-cell after 10 cycles (light purple dots), 100 cycles (dark purple dots), 200 cycles (green dots) and 400 cycles (blue dots). (**b**) Z real as function of w ^−1/2^ for 2D α-Ge 1 mg cm^−2^ semi-cell after 10 cycles (light purple dots), 100 cycles (dark purple dots), 200 cycles (green dots) and 400 cycles (blue dots). (**c**) Li^+^ diffusion coefficients (*D_Li_*^+^) (filled columns) and charge-transfer resistance (R_ct_) (dashed columns) during 400 cycles at constant current 0.5 C for 2D α-Ge 1 mg cm^−2^ semi-cell.

**Figure 5 nanomaterials-12-03760-f005:**
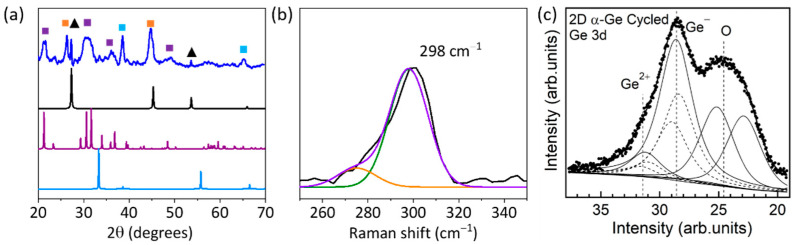
(**a**) PXRD pattern of 2D α-Ge anodes after the cycling process (blue line), simulated Li_2_CO_3_ (purple line), and simulated Li_2_O (blue line). Black triangles are assigned to Ge-crystalline, purple squares are assigned to Li_2_CO_3_, blue squares are assigned to Li_2_O, orange squares are assigned to VGCF, and green squares are assigned to Cu. (**b**) Raman spectrum of 2D α-Ge after the cycling process. Raman spectrum measurement (black line), deconvoluted contributions of crystalline α-Ge at 298 cm^−1^ (green line), amorphous Ge at 275 cm^−1^ (orange line), and cumulative plot of both contributions (purple), laser power of 1.75 mW. (**c**) Ge 3d and O 2s XPS peaks (hν = 1253.6 eV) for 2D α-Ge anode after the cycling process. Experimental data (dotted line) and resulting fit (thick continuous lines). Thin lines correspond to the different contributions and dashed lines to the spin-orbit splitting of Ge 3d.

**Table 1 nanomaterials-12-03760-t001:** Specific capacity (mAh g^−1^) at constant current rate 0.5 C.

Scan Rate 0.5 C (Specific Capacity)	Cycle 1	Cycle 20	Cycle 50	Cycle 100	Cycle 200	Cycle 400
2D α-Ge 1 mg cm^−2^	1104	884	711	697	608	599
2D α-Ge 2 mg cm^−2^	1526	1095	941	790	368	162

## Data Availability

Not applicable.

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
