# Peer review of "Alpha-Germanium Nanolayers for High-Performance Li-ion Batteries"

_nanomaterials, 2022, doi:10.3390/nano12213760_

Round 1
Reviewer 1 Report
This manuscript reports a method to construct gram-scale production of 2D α-Ge by LPE for high capacity lithium ion battery anode materials. Some minor issues should be addressed prior to published in Nanomaterials:
1.I think the units in Table 1 is messed up in Table 1 ,please double check.
2.To demonstrate the structural stability, the 2D α-Ge 1 electrode after electrochemical cycling is suggested to be provided.
Author Response
Reviewer 1
This manuscript reports a method to construct gram-scale production of 2D α-Ge by LPE for high capacity lithium ion battery anode materials. Some minor issues should be addressed prior to published in Nanomaterials:
Q1: I think the units in Table 1 is messed up in Table 1, please double check.
A1: Thank you to the reviewer for the appreciation. Accordingly, we have modified Table 1. We have kept the units only in the table caption for clarity. We have replaced the old Table 1 with the following:
Table 1. Specific capacity (mAh g-1) at constant current rate 0.5C.
|
Scan rate 0.5C (Specific capacity) |
Cycle 1
|
Cycle 20
|
Cycle 50
|
Cycle 100
|
Cycle 200
|
Cycle 400
|
|
|
|
|
|
|
2D α-Ge 1 mg cm-2 |
1104 |
884 |
711 |
697 |
608 |
599 |
|
|
|
|
|
|
2D α-Ge 2 mg cm-2 |
1526 |
1095 |
941 |
790 |
368 |
162 |
|
|
|
|
|
Q2: To demonstrate the structural stability, the 2D α-Ge 1 electrode after electrochemical cycling is suggested to be provided.
A2: We thank the reviewer for the comment. It is relevant to know the structural changes the electrode undergoes after the charge and discharge cycles. This information is beneficial to understand the chemical-physical processes involved during cycling. As indicated in the final section of the manuscript, to demonstrate the structural material stability after the cycling process, different measures were carried out, comparing the results to those of the pristine 2D α-Ge. XRPD, Raman Spectroscopy, and XPS. These experiments have allowed us to evaluate the structural changes. The XRPD analysis shows peaks at 2θ 27.20°, 45.24°, 53.64°, and 65.99° corresponding to crystalline 2D α-Ge. This indicates that the electrode retains some of its initial crystallinity. Also, Raman Spectroscopy corroborated that the electrodes keep between 75-80% of the 2D α-Ge crystalline after the cycling process. Finally, we also performed XPS to know the structural changes focused on the surface of the material. We found a relevant contribution of Ge-, corresponding to LixGe alloy associated with the charging process. The combined analysis using XRD, Raman, and XPS, shows high crystallinity retention of 2D α-Ge nanoflakes after continuous cycling.
Reviewer 2 Report
In this work, the authors reported the lithium storage performance of α-germanium (α- Ge) nanolayers which was prepared through wet ball-milling. Due to the two-dimensional feature of α-Ge, good electrochemical performance can be obtained. Generally, this manuscript is well-organized and result-oriented, and can be accepted after addressing the following issues.
1. Generally, the TEM and SEM images provided in Fig. 1-b can not confirm the wo-dimensional feature of α-Ge, and updated images should be provided.
2. For the electrodes with active material loading of 5 and 10 mg cm-2, are there any cracks can be detected on the electrode after drying?
3. For the calculations of ionic diffusion coefficient, the C in Equation (2) corresponds to the Na+ concentration in the bulk electrode, not the concentration of electrolyte. The author should revise this part.
4. The language of this manuscript should be improved.
Author Response
Q1. Generally, the TEM and SEM images provided in Fig. 1-b can not confirm the wo-dimensional feature of α-Ge, and updated images should be provided.
A1. TEM and SEM images typically cannot be used to measure the thickness of the nanolayers, while AFM does. Therefore, we also provided AFM images adding to the lateral dimensions and information on the thickness. In the new version of the SI, we have added a new image (Figure S3) showing the profile of the alpha-Ge nanolayer displayed in Fig. 1c to clarify the thickness of the nanolayers.
Q2. For the electrodes with active material loading of 5 and 10 mg cm-2, are there any cracks can be detected on the electrode after drying?
A2. The electrodes with higher active material loading present a similar aspect as electrodes with lower loading, without the presence of any cracks after drying and before assembling the battery.
Q3. For the calculations of ionic diffusion coefficient, the C in Equation (2) corresponds to the Na+ concentration in the bulk electrode, not the concentration of electrolyte. The author should revise this part.
A3. We thank the reviewer for the indication. We added the changes in the new version of the manuscript as “c is the concentration of Li+ in the bulk electrode“.
Q4. The language of this manuscript should be improved.
A4. Thank you to the reviewer for this suggestion. We revised the manuscript to correct the grammatical mistakes.
Reviewer 3 Report
The paper reports a study of 2D alpa-germanium nanolayers as electrodes for LIBs. The study is soundly designed, sufficient analysis is performed, the introduction and discussion are very nice and comprehensive. The paper is written in good English. Supplementary data complements the study.
In my opinion, the paper requires some minor changes before publication:
1) Paragraphs 96-100 and 135-140 are duplicates, one of them must be deleted.
2) In Fig.1 the numbers must be put on the scale, and not in the description of the SEM image.
3) In line 255 it is not obvious, whether the 43% loss is observed after the first 50 cycles, or after 200 cycles. Please, rephrase.
4) In the paragraph 264-273 the need for multiple additional expariments with higher loading of Ge is not clear, since the literature data is availiable. On the cocntrary, it might be interesting to see an experiment with slightly smaller loading.
5) It might be good to separate Fig. 2 into 2 figures and fut them closer to where these figures are mentioned (after lines 236, for example). Sasme applies to Fig.3 - it might be moved up in the text.
6) In line 296 there is a typo, it must be "c) GCD at different scan rates"
I believe that these small improvements will help perfect the article and then it will be published with no delay.
I would like to thank the authors for such good work, it was my pleasure to review it.
Author Response
Q1. Paragraphs 96-100 and 135-140 are duplicates, one of them must be deleted.
A1. We thank the reviewer for the indication. In the new version of the manuscript, we have deleted paragraphs 96-100 in the “Results and Discussion” section and kept them in the “Materials and methods” section.
Q2. In Fig.1 the numbers must be put on the scale, and not in the description of the SEM image.
A2. We thank the reviewer for this suggestion. We have added the numbers on the scale in Fig.1 in the new version of the manuscript.
Q3. In line 255 it is not obvious, whether the 43% loss is observed after the first 50 cycles, or after 200 cycles. Please, rephrase.
A3. We thank the reviewer for this indication. We have revised the corresponding paragraph and rephrased it with: “Also, over 200 cycles, the electrode with 2 mg cm-2 shows a dramatic capacity loss of 75% compared to the initial value, meanwhile for 1 mg cm-2 only loss 45% compared to the initial value“.
Q4. In the paragraph 264-273 the need for multiple additional expariments with higher loading of Ge is not clear, since the literature data is availiable. On the cocntrary, it might be interesting to see an experiment with slightly smaller loading.
A4. We thank the reviewer for this pertinent comment. The experiments with high mass loading were performed to obtain the limitation of mass loading with the same geometrical area to obtain higher capacity if possible. Nevertheless, the worst performance at high mass loading can be attributed to the swelling effect and loss of active area due to the increased thickness. Considering the capacity obtained, we believe that a minimum of 1 mg cm-2 is necessary with this type of material.
Q5. It might be good to separate Fig. 2 into 2 figures and fut them closer to where these figures are mentioned (after lines 236, for example). Sasme applies to Fig.3 - it might be moved up in the text.
A5. In the new version of the manuscript, we split Fig 2 into two new figures (Fig 2 and Fig 3), and both have been moved up in the text. We followed the same with Fig. 3, now called Fig.4, in the new version of the manuscript.
Q6. In line 296 there is a typo, it must be "c) GCD at different scan rates"
A6. We corrected the mistake in the revised manuscript.